# MODIS Land Surface Temperature Product Reconstruction Based on the SSA-BiLSTM Model

**Jianyong Cui [1], Manyu Zhang [1], Dongmei Song [1,2,*], Xinjian Shan [3] and Bin Wang [1]**

1    College of Oceanography and Space Informatics, China University of Petroleum (East China), Qingdao 266580, China; cui_jianyong@upc.edu.cn (J.C.); z20160093@s.upc.edu.cn (M.Z.); wangbin007@upc.edu.cn (B.W.)
2    Laboratory for Marine Mineral Resources, Qingdao National Laboratory for Marine Science and Technology, Qingdao 266071, China
3    Institute of Geology, China Earthquake Administration, Beijing 100029, China; xjshan@ies.ac.cn
*    Correspondence: songdongmei@upc.edu.cn

**Abstract:** Land surface temperature (LST) is an important parameter indispensable for studying the substance and energy exchanges between the land surface and the atmosphere, climate changes, and other related aspects. However, due to cloud cover, there are many null values in MODIS (Moderate Resolution Imaging Spectroradiometer) LST data, which prevents such data from being widely used. Therefore, an LST reconstruction method is proposed by combining data decomposition with data prediction—SSA (Singular Spectrum Analysis) and BiLSTM (Bidirectional Long Short-Term Memory). This method consists of two major processes, namely, rough LST reconstruction based on the SSA model and refined LST reconstruction based on the BiLSTM model. The accuracy of the proposed method is verified through "removal–reconstruction–comparison" using remote sensing data and measured data. The verification results show that when the rate of original missing values in the LST time series for the study area is lower than 10%, the RMSE is smaller than 1.1 K, and the correlation coefficient is more significant than 0.98. Even when the rate of missing data is 40% and 50%, the proposed method remains accurate, the values of RMSE are 1.8331 K and 2.2929 K, and the importance of $R^2$ are 0.9856 and 0.9800, respectively. The proposed method is compared with other existing LST reconstruction methods. The results of the comparative analysis indicate that the proposed method is superior to other methods in terms of reconstruction accuracy and stability. Moreover, the LST data reconstructed using the proposed method are highly consistent with the measured data, which further proves the accuracy of this method in LST reconstruction. The research findings provide a new technique and idea for accurate LST reconstruction.

**Keywords:** land surface temperature (LST); SSA; BiLSTM; reconstruction; MODIS; data decomposition

## 1. Introduction

Land surface temperature (LST) is an essential parameter for studying surface energy balance and land surface processes [1] and a key factor relevant to climate changes, vegetation, and ecological monitoring of cities. It plays a vital role for research on global climate changes. MODIS (Moderate Resolution Imaging Spectroradiometer) data have gradually become an important means to obtain LST due to its comprehensive coverage and long observation period. However, the space-time continuity of MODIS LST data may be seriously impaired by clouds and cloud shadows. In 2019, Mao et al. [2] found in their research that about 65% of the world's land surface was always covered by clouds, resulting in a large number of missing values in thermal infrared remote sensing images, and the specific number of missing values varied by region. This problem seriously affects the wide use of MODIS LST data. Therefore, LST reconstruction is a precondition for the effective use of LST data in research on climate changes, urban heat islands, and other related aspects.

In recent years, LST reconstruction methods have attracted wider attention from scholars. A series of research achievements have been made in the last two decades. These LST reconstruction methods can be classified into three categories. 1. LST reconstruction methods based on spatial-domain information [3]. These methods perform interpolation based on the spatial correlation between missing pixels and adjacent clear-sky pixels and include the spline function method [4], regression tree analysis method [5], and Kriging [3]. Such methods do not need other auxiliary information and are easy to realize, but they have certain deficiencies, such as the lack of clarity and precision of reconstructed images. For this reason, these methods are applicable only when there are a small number of missing pixels in the spatial domain. 2. LST reconstruction methods based on time-domain information [6]. These methods reconstruct the missing pixels based on LST changes on the same time axis. Such methods mainly include harmonic analysis method [7], multi-temporal robust regression method [8], singular spectrum analysis method [9], daily temperature cycle model [6], physical modeling [10], and SG (Savitzky Golay Filter) method [11]. For these methods, the reconstruction of LST based on the time series decomposition algorithm mainly involves two strategies. One strategy is to obtain several different subseries with varying cycles, characteristics, and rules of change through data decomposition, perform prediction for each subseries, and finally find the sum of predicted results to obtain reconstructed data. The other strategy is to perform data decomposition, remove the residuals, keep the subseries containing information on the trend of changes in data, and then find the sum of the values simultaneously in these subseries to obtain the interpolated values. However, these methods perform interpolation based on the trend of changes in LST in the time domain. As a result, the smoothing of data series will be inevitable, resulting in the loss of abruptly changing LST information. Therefore, these methods are applicable only when there are a small number of missing pixels in the time domain. 3. LST reconstruction methods are based on the information in both the spatial and time domains [12–14]. These methods achieve data reconstruction first performing interpolation in the time domain and then in the spatial domain. Because the information in both the time domain and the spatial domain is used at the same time, these methods can accurately reconstruct the data in areas with many missing pixels. Still, they are greatly affected by the high heterogeneity of LST over space and time.

In summary, when there are a large number of missing values in LST data, traditional LST reconstruction methods will no longer be applicable, and the accuracy of data reconstructed with traditional methods will be insufficient to meet the requirements of practical application. In recent years, LST reconstruction methods based on deep learning have been used in LST reconstruction to solve the problem mentioned above [15–17]. Such methods are characterized by strong learning ability, high robustness, and no need for complex models with clear catalytic expressions and fully consider the heterogeneity of LST over space and time. The models are built based on the relationship between LST and environmental variables for most of these methods. Therefore, the accuracy of the models is greatly affected by the number and type of training samples, and the features contained in the time series data are neglected. For this reason, the accuracy of reconstructed data cannot meet the needs of practical application. The research findings of research in recent years indicate that hybrid models combining data decomposition models with certain predictive models perform better than ordinary models in prediction [18]. Hybrid models have been widely used in various fields. Compared with the Empirical Mode Decomposition (EMD) method and other data decomposition algorithms [19], the Singular Spectrum Analysis (SSA) method can identify the potential cycle and trend features of data more adequately and obtain more abundant data features. Compared with other predictive algorithms based on deep learning, the Bidirectional Long Short-Term Memory (BiLSTM) network can better learn the short-term features in the entire time series, thus preventing the abruptly changing information from being smoothed easily and delivering more accurate prediction results.

In this paper, an LST reconstruction method combining data decomposition and data prediction is proposed—SSA-BiLSTM. This method firstly performs rough LST data

reconstruction by extracting the long-term features and change trends of the data using the SSA model and then complete refined LST data reconstruction by learning the short-term features of the data using the BiLSTM model. Experimental results prove the proposed method's good performance and high robustness in LST reconstruction.

In Section 2, the products and data used for analysis and the works related to data preprocessing are described. In Section 3, the basic principles of SSA and BiLSTM are introduced, and the reconstruction method based on the SSA-BiLSTM model is described in detail. In Section 4, the accuracy of the proposed method in LST reconstruction is analyzed qualitatively and quantitatively using remote sensing data and measured data. The advantages and disadvantages of the proposed method are summarized in Section 5.

## 2. Study Area and Data

### 2.1. Study Area

The study area is located in the Hotan region in the southern part of Xinjiang, Northwest China. The center of the study area is at 79.92°E, 37.12°N. The terrain is low in the north and high in the south. The study area borders the Tarim Basin in the north and the Kunlun Mountains in the south. Gobi deserts and oases are widely distributed in the part of the Hotan region to the north of the Kunlun Mountains. The significant types of landforms of this area are mountains and basins. The area of mountains accounts for 33.3%, the area of Gobi deserts accounts for 63%, and that of oases accounts for only 3.7%. This area has a typical continental climate dominated by abundant heat and very little precipitation throughout the year. It is one of the regions in China where sandstorms are most common. Steam and clouds have little impact on the sensors during the scanning of this area, and the quality of LST data collected by the sensors is relatively high. Therefore, selecting this area as the study area makes it possible to better evaluate the accuracy and reliability of the LST reconstruction method proposed in this paper.

### 2.2. Data and Data Preprocessing

#### 2.2.1. Research Data

In this paper, the MODIS MYD11A2 LST 8-day composite data products at 1 km resolution are used as the research data. Since the missing amount of MODIS 8-day synthetic data are smaller than that of daily data, the reconstruction experiment using 8-day synthetic data can better evaluate the accuracy of the proposed method. LST reconstruction experiments were conducted using the LST data collected at 1:30 a.m. from 2015 to 2020. The data products were obtained by taking an average of the LST data of eight days under clear sky conditions. The daily LST data were collected by the MODIS sensor onboard the Aqua satellite. The MODIS sensor can provide global LST data with high observation time frequency and wide spatial coverage. In addition, the effectiveness of the proposed method was verified using the LST data measured at Yutian Station, Cele Station, Hotan Station, Luopu Station, Moyu Station, and Pishan Station in the study area (the locations of these weather stations are shown in Figure 1) in 2020. The measured data are obtained hourly from the Forain Weather System, with a measurement error of 0.1 K.

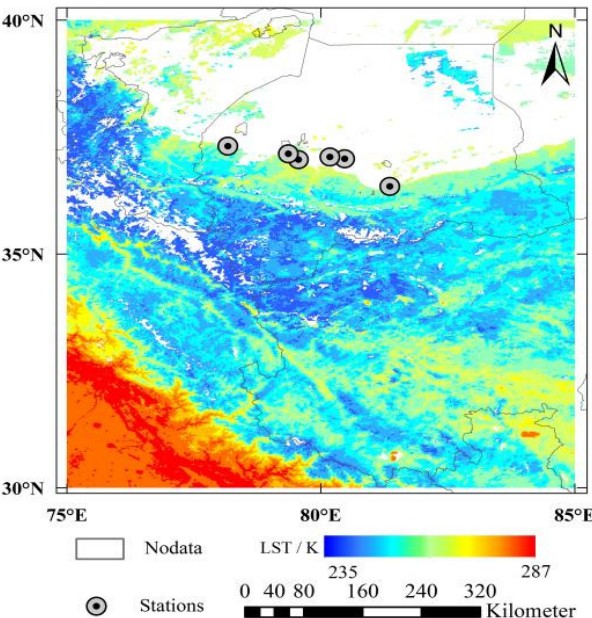

**Figure 1.** Chart of MODIS LST in the study area as of April 2008.

### 2.2.2. Data Preprocessing

The MODIS data products used for this study are HDF files. The original images need to be stitched, cropped and re-projected using the MRT (MODIS Projection Tool) provided by NASA. A $10° \times 10°$ area of the stitched LST data after midnight with Hotan, Xinjinag as the center point was selected, cropped, and re-projected as Geographic Lat/Lon. For measured data at weather stations, the average of data measured at 1:00 a.m. and 2:00 a.m. was taken as the measured data at 1:30 a.m., the 8-day mean value was calculated, and then the data consistent with the time of MYD11A2 were generated for verification.

## 3. LST Reconstruction Method

In this paper, an LST reconstruction method combining data decomposition and data prediction is proposed—SSA-BiLSTM. This method consists of two major processes, namely, rough LST reconstruction based on the trend features of the data extracted using the SSA model and refined LST reconstruction based on the short-term features of the data learned by BiLSTM model. The detailed process of the proposed method is shown in Figure 2.

### *3.1. Rough LST Reconstruction Based on the SSA Model*

#### 3.1.1. SSA Algorithm

SSA is a powerful method that has emerged in recent years to study nonlinear time series data features. It converts the observed time series data into a trajectory matrix, decomposes and reconstructs the trajectory matrix, and extracts various signals representing the different components of the original time series, such as long-term trend signals, periodic signals and noise signals, and then completes the grouping and reconstruction of the time series data.

The basic idea of SSA is to convert the observed one-dimensional time series data into its trajectory matrix:

$$X = \left(x_{ij}\right)_{i,j=1}^{L,K} = \begin{pmatrix} y_1 & y_2 & \cdots & y_K \\ y_2 & y_3 & \cdots & y_{K+1} \\ \vdots & \vdots & \cdots & \vdots \\ y_L & y_{L+1} & \cdots & y_T \end{pmatrix} \qquad (1)$$

where T stands for time, L is the selected window length, and $K = T - L + 1$. $XX^T$ is calculated, and then singular value decomposition is performed to obtain an L number of characteristic values and the corresponding feature vectors. The new time series is reconstructed by analyzing and combining the signal represented by each characteristic value. In a nutshell, SSA can be divided into four processes, namely, the construction of trajectory matrix, SVD, grouping, and reconstruction. The purpose of collection is to separate the target signals from other signals and thereby create a new target series.

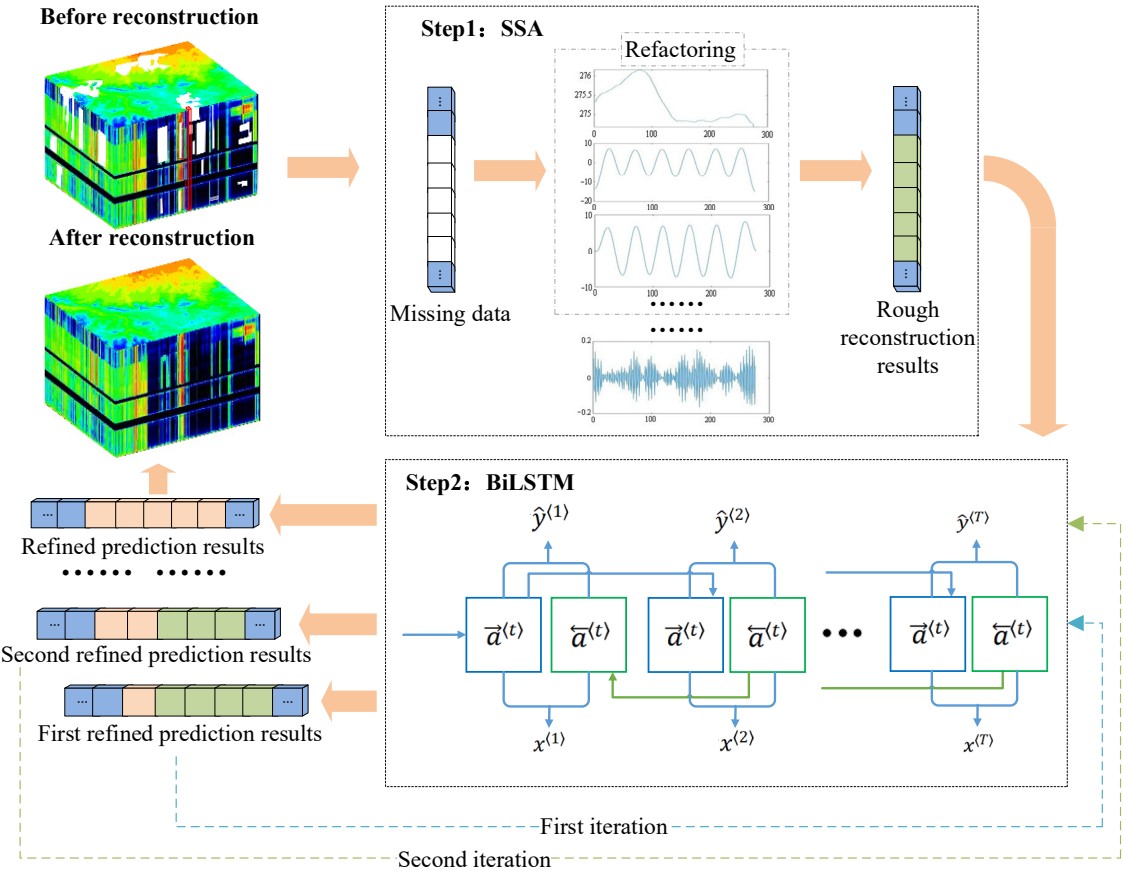

**Figure 2.** Process diagram of LST reconstruction based on the SSA-BiLSTM model.

The data product used for this study has 46 values in one year. Therefore, the window length of the SSA model is set to 46. After the time series of each pixel is decomposed using the SSA model, a $46 \times 46$ two-dimensional subseries matrix can be obtained. Each subseries has different data features, and each subseries consists of 46 values. When the data are complete without defect, SSA can better decompose the long-period and trend characteristics of the data. The more subseries used for reconstruction, the higher the correlation between the reconstructed data and the original data. However, in practice, there are a large number of missing values in the land surface temperature time series, and the subseries decomposed by SSA will be affected by the missing value information. At this time, the number of subsequences used for data reconstruction should be selected according to the correlation between the reconstructed data and the original data. In this paper, the experiment is carried out. Two groups of complete surface temperature time series: data1 and data2 are selected in the study area to produce 10%, 20%, 30%, 40%, and 50% data loss rates, respectively, and ensure that the data loss time and value continuity of the two groups of data in the time domain are different. The optimal number of subseries is found by analyzing the correlation between the reconstructed data composed of the addition of the number of different subseries and the original data. The results are shown in Table 1. It can be seen from the table that the selection of the number of optimal subseries

of the same data is also different due to the different amount of data missing. In addition, it can also be seen that due to the difference of missing values of different pixels in the time domain, the selection of the number of optimal subseries of different data is also different. Therefore, this algorithm needs to decompose each pixel one by one and determine the optimal number of subseries according to the principle of the highest correlation. Hence, the SSA model can effectively extract the trend features of the original data, thus making rough LST reconstruction (i.e., the first interpolation) possible. In summary, the selection of the number of subseries used for reconstruction can be summarized as follows:

$$Z_n = \sum_{i=1}^{n}(z_{1+}z_2 + \cdots + z_n) \cdots \cdots (n = 1:46) \tag{2}$$

$$D_{opt} = max\left[r^2(Z_n, D_{ini})\right] \cdots \cdots (n = 1:46) \tag{3}$$

where, $Z_n$ represents the reconstructed data composed by adding different numbers of subseries, $z_n$ represents different subseries decomposed by SSA, $n$ represents the number of decomposed subseries, $D_{opt}$ represents the optimal reconstructed data, and $D_{ini}$. represents the original data.

**Table 1.** Correlation between different reconstruction items and the original series ($R^2$).

| Rate of Missing Values | | First 3 Items | First 5 Items | First 7 Items | First 9 Items | First 11 Items | First 13 Items | First 15 Items | First 20 Items |
|---|---|---|---|---|---|---|---|---|---|
| 10% | data1 | 0.9483 | 0.9800 | 0.9870 | 0.9923 | 0.9966 | 0.9977 | 0.9979 | 0.9979 |
| | data2 | 0.9490 | 0.9800 | 0.9867 | 0.9919 | 0.9972 | 0.9983 | 0.9987 | 0.9987 |
| 20% | data1 | 0.9485 | 0.9800 | 0.9867 | 0.9917 | 0.9961 | 0.9972 | 0.9972 | 0.9972 |
| | data2 | 0.9489 | 0.9796 | 0.9862 | 0.9914 | 0.9961 | 0.9972 | 0.9975 | 0.9975 |
| 30% | data1 | 0.9484 | 0.9788 | 0.9857 | 0.9903 | 0.9954 | 0.9963 | 0.9963 | 0.9963 |
| | data2 | 0.9478 | 0.9778 | 0.9842 | 0.9883 | 0.9926 | 0.9933 | 0.9931 | 0.9927 |
| 40% | data1 | 0.9291 | 0.9162 | 0.9340 | 0.9356 | 0.9378 | 0.9365 | 0.9362 | 0.9354 |
| | data2 | 0.9466 | 0.9775 | 0.9834 | 0.9870 | 0.9904 | 0.9910 | 0.9907 | 0.9900 |
| 50% | data1 | 0.8925 | 0.8199 | 0.8124 | 0.8196 | 0.8174 | 0.8199 | 0.8199 | 0.8181 |
| | data2 | 0.9383 | 0.9491 | 0.9403 | 0.9397 | 0.9326 | 0.9325 | 0.9319 | 0.9319 |

### 3.1.2. Rough LST Reconstruction Method Based on the SSA Model

The process of rough LST reconstruction based on the SSA model is shown in Figure 2. The detailed process is given below. First, the time series LST data with missing pixels are decomposed using the SSA model to obtain subseries data containing different features. Second, the correlation between the sums of different numbers of subseries data and the original LST data is analyzed, and the most correlative results are selected and used as the data for reconstruction. Due to the difference in the number of missing values for different pixels in the time domain, this algorithm needs to perform SSA decomposition of each pixel and determine the optimal number of subseries based on the principle of highest correlation. Finally, the original zero-value data in the missing pixels are replaced with the reconstructed data and rough LST reconstruction is completed to provide input data for refined LST reconstruction based on the BiLSTM model.

### 3.2. Refined LST Reconstruction Based on the BiLSTM Model
### 3.2.1. Principle of the BiLSTM Model

The BiLSTM model has been developed based on LSTM networks, including forward-pass and backward-pass LSTM networks. LSTM networks are a recurrent neural network proposed by Hochreiter and Schmidhube [20] in 1997 to over the vanishing gradient problem in RNNs. Some neurons in the hidden layers of RNNs are replaced with LSTM neurons, effectively solving the vanishing gradient problem in traditional RNNs. An LSTM

network includes "three gates", namely, the forget, input, and output gates. The "gate" structure and "cell status" selectively allow the obtained information to act on the status in the RNN at each point in time. The "forget", "update", and "output" mechanism enables the entire network to determine in a more effective way which information should be kept and which information should be forgotten [17]. The structure of the LSTM neural network is shown in Figure 3.

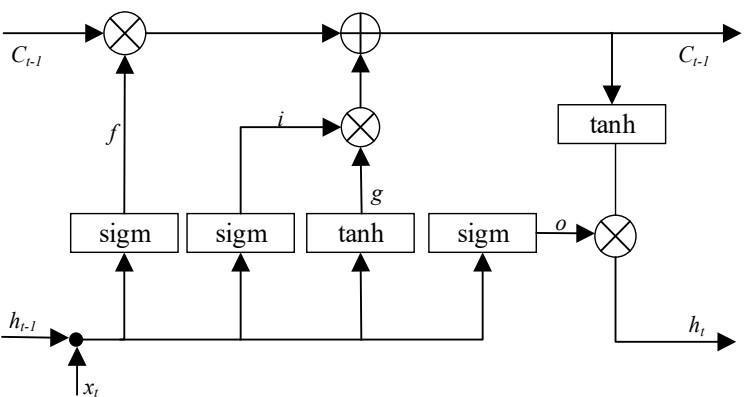

**Figure 3.** Structure of the LSTM neural network.

In Figure 4, *i* is the input gate, *o* is the output gate, *f* is the forget gate, *g* is the hidden status, *ct* is the cell status at time *t*, and the blocks represent the neural network layers. The Equations below are used to calculate the parameters of various gates:

$$f_t = \sigma\left(W_f \times [C_{t-1}, h_{t-1}, x_t] + b_f\right) \tag{4}$$

$$i_t = \sigma(W_i \times [C_{t-1}, h_{t-1}, x_t] + b_i) \tag{5}$$

$$O_t = \sigma(W_o \times [C_{t-1}, h_{t-1}, x_t] + b_o) \tag{6}$$

$$C_t = f_t \times C_{t-1} + (1 - f_t) * Ct \tag{7}$$

$$h_t = o_t * tanh(C_t) \tag{8}$$

where *W* is the weight matrix, *b* is the bias matrix, subscripts "*i, f, o*, and *g*" denote the *i, f,* and *o* gates and the hidden status, and $\sigma$ is the neural network's activation function. The volume of data transmitted from the previous time to the current time can be obtained through the forget gate, and the volume of data transmitted from the current time to the subsequent time can be obtained through the output gate. *g* can be obtained from the input value at the current time and the status at the previous time.

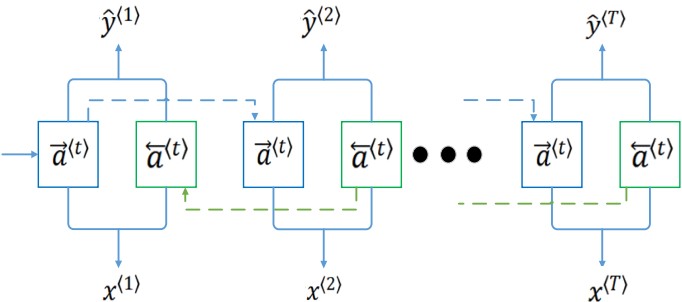

**Figure 4.** Diagram of the BiLSTM model structure.

Due to its ability to read data bidirectionally, the model BiLSTM can better capture the features of data and information, thus overcoming the deficiency of LSTM networks that can only acquire information from the forward direction and cannot learn future data features. Bidirectional recurrent neural networks can produce more accurate prediction

results because they use the information in the past and in the future at the same time and perform network training in both positive and negative directions. The network structure of the BiLSTM model used for this study is shown in Figure 4. In this model, the output process of forward-propagation neurons is independent from that of backward-propagation neurons (the calculations are done using Equations (9)–(11)).

$$\overrightarrow{a}^{\langle t \rangle} = \varphi_1(W_{\overrightarrow{a}}[x^{\langle t \rangle}; a^{\langle t-1 \rangle}] + b_{\overrightarrow{a}}) \tag{9}$$

$$\overleftarrow{a}^{\langle t \rangle} = \varphi_2(W_{\overleftarrow{a}}[x^{\langle t \rangle}; a^{\langle t+1 \rangle}] + b_{\overleftarrow{a}}) \tag{10}$$

$$\hat{y}^{\langle t \rangle} = \varphi_2(W_y[\overrightarrow{a}^{\langle t \rangle}; \overleftarrow{a}^t] + b_y) \tag{11}$$

where $\varphi$ represents the activation function, $b$ represents the offset term, $W$ represents the weight coefficient, $t$ represents the time, $\hat{y}$ represents the final output, $\overrightarrow{a}^{\langle t \rangle}$ and $\overleftarrow{a}^{\langle t \rangle}$ denote the results of calculation in the positive and negative directions. However, the BiLSTM model needs to obtain the entire input series before predicting. As mentioned earlier, the missing data are roughly reconstructed using the SSA model, which preliminarily meets the requirements of the BiLSTM model for data completeness.

### 3.2.2. Refined LST Reconstruction Method Based on the BiLSTM Model

The process of refined LST reconstruction based on the BiLSTM model is shown in Figure 2. The detailed process is given below. The roughly reconstructed LST data are input into the standard BiLSTM model, the first 90% of the data are selected and used for training, and the last 10% of the data are used for testing. BiLSTM initialization parameter settings are shown in Table 2. Secondly, based on the BiLSTM model, the iterative prediction method is used for the fine reconstruction of surface temperature. In the time series, the surface temperature of the missing point is predicted according to the sequence of the missing time. Every time the surface temperature data of a missing point are predicted and the input data are updated, it is called an iteration. If there are n missing points in the time series, it needs to be iterated n times to realize the fine reconstruction of the surface temperature. The specific implementation process is as follows: Depending on the sequence of time of missing data, the trained BiLSTM model is used to predict the LST at the first missing data point, the rough reconstruction result corresponding to the first missing data point is replaced with the prediction result, and complete the updating of input data. At this point, the first iteration of the refined LST reconstruction process is complete. Then, the result of the first iteration is used as the input data to predict the LST at the second missing data point, and the original rough reconstruction result is replaced with the prediction result. At this point, the second iteration of the refined LST reconstruction process is complete. Iterations and predictions are performed repeatedly according to the procedure described above until the refined reconstruction of all the missing pixels in the time series is completed. In the model training process, the model parameters are optimized continuously depending on the reconstruction accuracy of the test sets to increase the accuracy of reconstruction results.

**Table 2.** BiLSTM initialization parameter setting.

| Number of LSTM Layers | Number of Training Cycles | Number of Nodes in Hidden Layers | Learning Rate | Ratio of Input Parameters to Output Parameters |
|:---:|:---:|:---:|:---:|:---:|
| 2 | 100 | 20 | 0.005 | 10:1 |

### 3.3. Evaluation Criteria

Three typical evaluation indices, namely, root mean square error (RMSE), mean absolute percentage error (MAPE), and correlation coefficient ($R^2$), are used to evaluate recon-

struction accuracy. The performance of various methods in LST reconstruction is evaluated by comparing the values after reconstruction with the original values. Equations (12)–(14) are the mathematical expressions of the three evaluation indices.

(1)    Root Mean Square Error (RMSE):

$$RMSE = \sqrt{\frac{1}{N} \sum_{i=1}^{N} \left( \hat{h}(i) - h(i) \right)^2} \tag{12}$$

(2)    Mean Absolute Percentage Error (MAPE):

$$MAPE = \frac{1}{N} \sum_{i=1}^{N} \frac{\left| \hat{h}(i) - h(i) \right|}{h(i)} \times 100\% \tag{13}$$

(3)    Correlation Coefficient ($R^2$):

$$R^2 = \frac{\sum_{i=1}^{N} \left( \hat{h}(i) - \bar{\hat{h}} \right) \left( h(i) - \bar{h} \right)}{\sqrt{\sum_{i=1}^{N} \left( \hat{h}(i) - \bar{\hat{h}} \right)^2} \sqrt{\sum_{i=1}^{N} \left( h(i) - \bar{h} \right)^2}} \tag{14}$$

where $N$ is the total number of missing values in LST time series data, $h(i)$ is the original value, $\hat{h}(i)$ is the LST after reconstruction, $\bar{h}$ is the average value of the original LST, and $\bar{\hat{h}}$ is the average value of the reconstructed LST.

## 4. Results and Discussion

The accuracy of the method proposed in this paper was analyzed quantitatively and qualitatively using remote sensing data and measured data. In addition, the proposed method was compared with other three LST reconstruction methods, including the LST reconstruction method based on SSA, the LST reconstruction method based on SG filter, and the LST reconstruction method based on SSA-LSTM. The LST reconstruction method based on SSA relies on data decomposition and iterative prediction for LST reconstruction. The LST reconstruction method based on SG filter performs least squares data fitting using higher order polynomials and completes data reconstruction through a weighing filter. The only difference between the third and proposed method is in the size of predictive models used. The verification results indicate that the prediction results produced by the BiLSTM model are more accurate than those produced by the LSTM model.

### 4.1. Quantitative Analysis

Firstly, a comparative analysis was performed using the "removal–reconstruction–comparison" process to analyze the performance of various methods in LST reconstruction involving varying rates of missing data. The principle of this analysis method is to remove some existing data randomly from the complete time series, reconstruct the missing data using different reconstruction methods, and compare the original values of missing pixels with reconstructed data. Therefore, the LST data with 500 pixels in six consecutive years in the study area were randomly selected, some existing data were removed randomly to achieve missing rates of 10%, 20%, 30%, 40%, and 50%, and the results of LST reconstruction using the methods above were analyzed statistically (The statistical results of average accuracy are shown in Figure 5). It can be seen from Figure 5 that the proposed method is superior to other methods in terms of overall accuracy in LST reconstruction at varying missing rates. For the proposed method, the maximum coefficient of correlation between the original values of missing data points and reconstructed data is 0.9942, the minimum value of RMSE is 1.1069, and the minimum value of MAPE is 0.3210. The method based on SG filter has the lowest accuracy in LST reconstruction, its reconstruction error at 50% missing rate is more significant than 4 K, and its reconstruction accuracy is 2.4 K lower than that of the proposed method. In addition, a group of data are randomly selected

from the above 500 pixels to analyze the correlation before and after reconstruction, so as to more clearly show the difference between the reconstructed LST data and the original value. The analysis results are shown in Figure 6. It can be seen from figure that when the missing rate is high, compared to other methods, the correlation between the LST data reconstructed with the proposed method and the original data is the highest. In comparison, the correlation between the LST data reconstructed the LST reconstruction method based on SG filter and the original data is the lowest, and there are great differences at some missing data points before and after reconstruction.

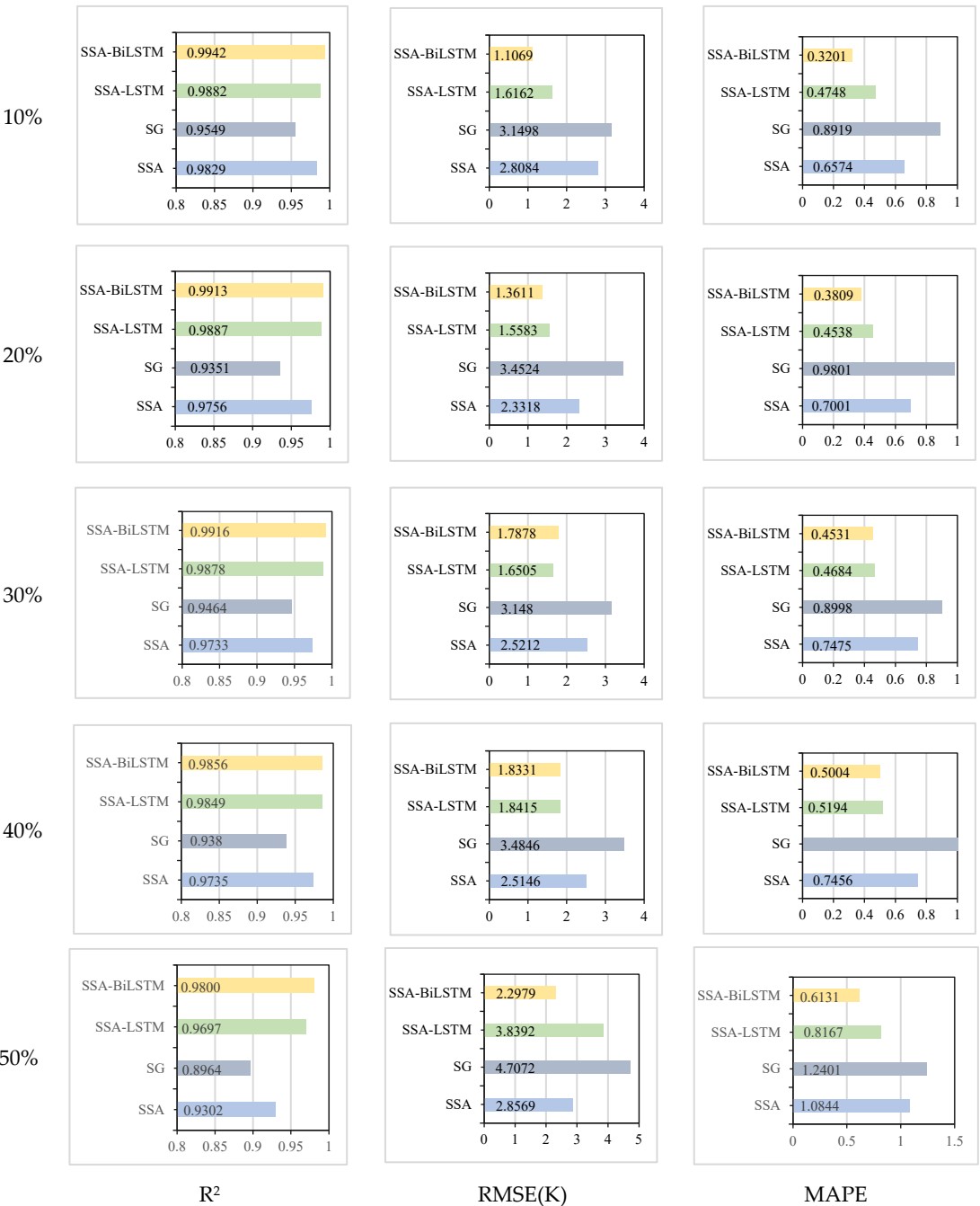

**Figure 5.** Comparison of the accuracy levels of different LST reconstruction methods at varying missing rates.

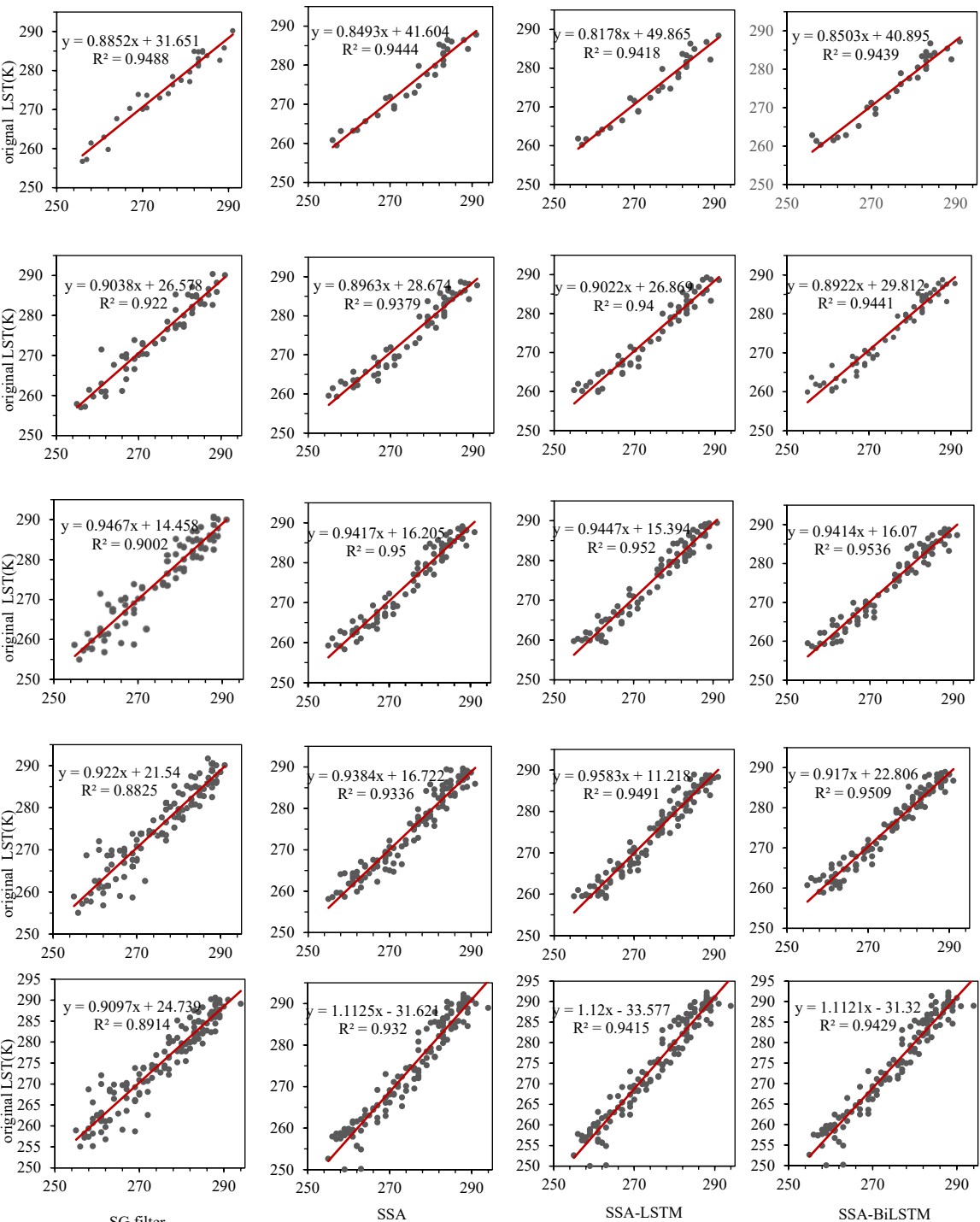

**Figure 6.** Comparative analysis of the correlation between LST data reconstructed with different methods and the original data at varying missing rates. The figures in lines 1 to 5 show the correlation analysis results of different LST reconstruction methods when the data missing rate is 10%, 20%, 30%, 40% and 50%, respectively.

The accuracy of the method proposed in this paper was further verified using the data measured at a number of weather stations. In Section 2.2.2, the time consistency between the measured data of the meteorological station and the surface temperature data of MYD11A2 has been processed. Therefore, the measured data used in this experiment have 46 values every year. Firstly, 40% of the MODIS LST time series data were removed in 2020 at Yutian Station, Cele Station, Hotan Station, Luopu Station, Moyu Station, and Pishan

Station, ensuring consistency of the times of missing values in each group of data. Then, LST data reconstruction was performed using the proposed method and other methods. There were 96 missing values in the data measured in 2020 at the six weather stations. The data measured at these weather stations corresponding to the times of missing values in 2020 were used to verify the accuracy of the proposed method. The correlation between the LST data reconstructed with this method and measured data were analyzed. The analysis results are shown in Figure 7. It can be seen from Figure 7 that the coefficient of correlation between the reconstructed data of the method in this paper and the data measured at weather stations is 0.9108, while the coefficient of correlation between the original data and measured data is 0.9231. These two values are basically consistent. In addition, it can be seen from the scatter plots of the original MODIS LST data measured after midnight and the reconstructed LST data with the proposed method that most data points before and after reconstruction are concentrated near the 1:1 line, which further proves the high accuracy of the proposed method.

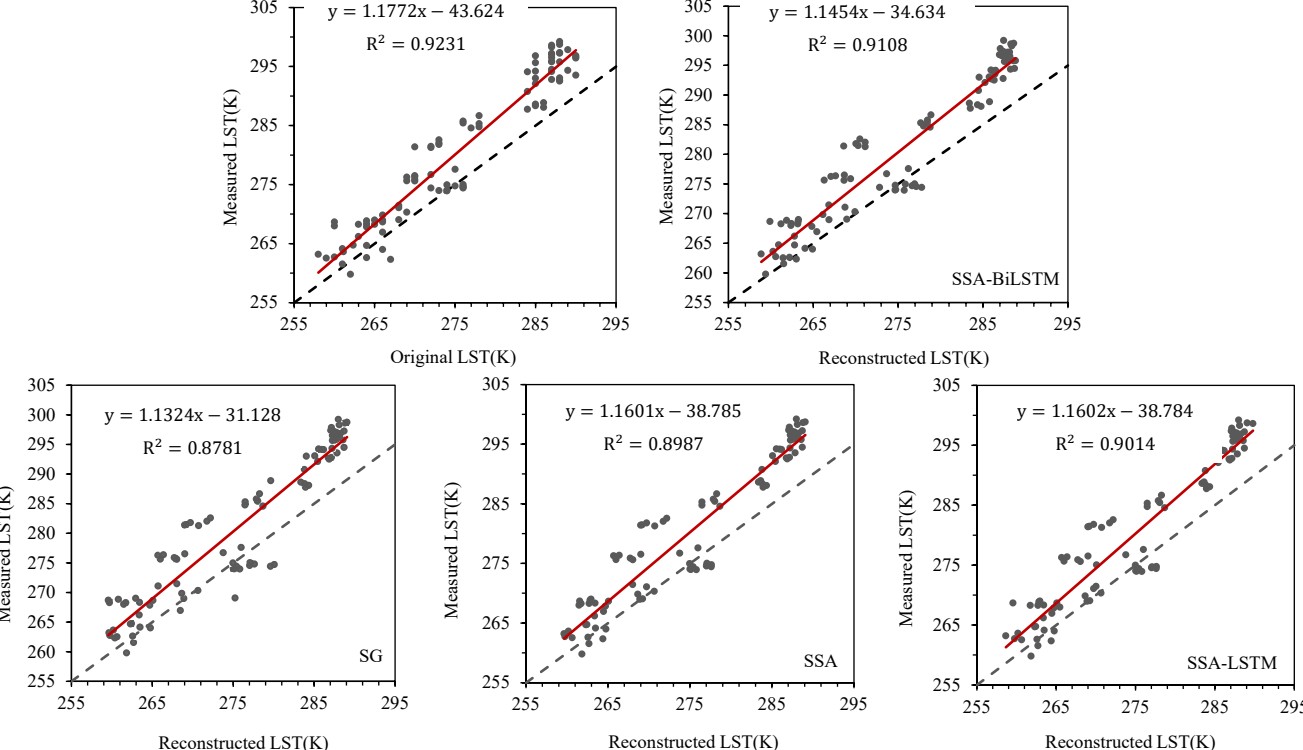

**Figure 7.** Comparison between LST data after midnight and measured data before (plot on the left side) and after (plot on the right side) reconstruction.

### 4.2. Qualitative Analysis

In order to evaluate the performance of various LST reconstruction methods in a more visual way, the complete LST data of year 2020 within $31 \times 31$ pixel range were selected in the study area shown in Figure 1 and used as the data for the experiments. Some existing data were removed randomly to create a missing rate of 40%. The areas with missing data were reconstructed using the four methods mentioned above. Figure 8 shows the reconstruction effects of different methods when the surface temperature data are continuously missing in the time domain. It can be seen from Figure 8 that the values of LST data reconstructed using the method based on SG filter are slightly higher. In practice, due to the influence of weather or external environment, there are some abrupt changes in the time series of surface temperature, such as the sudden drop of temperature. SG reconstruction method mainly uses the data before and after the missing point to reconstruct the missing pixel. When the data value before and after the missing point

are large, the value reconstructed using SG will be higher than the original result. In addition, the reconstruction method based on SSA lacks the details of time series due to the complement based on the trend characteristics of the data, so the reconstruction results are not accurate enough. The images reconstructed using the two methods based on SSA-LSTM and SSA-BiLSTM are more consistent with the original images. Since the BiLSTM model can read data bidirectionally, it can learn more potential data features and predict results more accurately than SSA-LSTM.

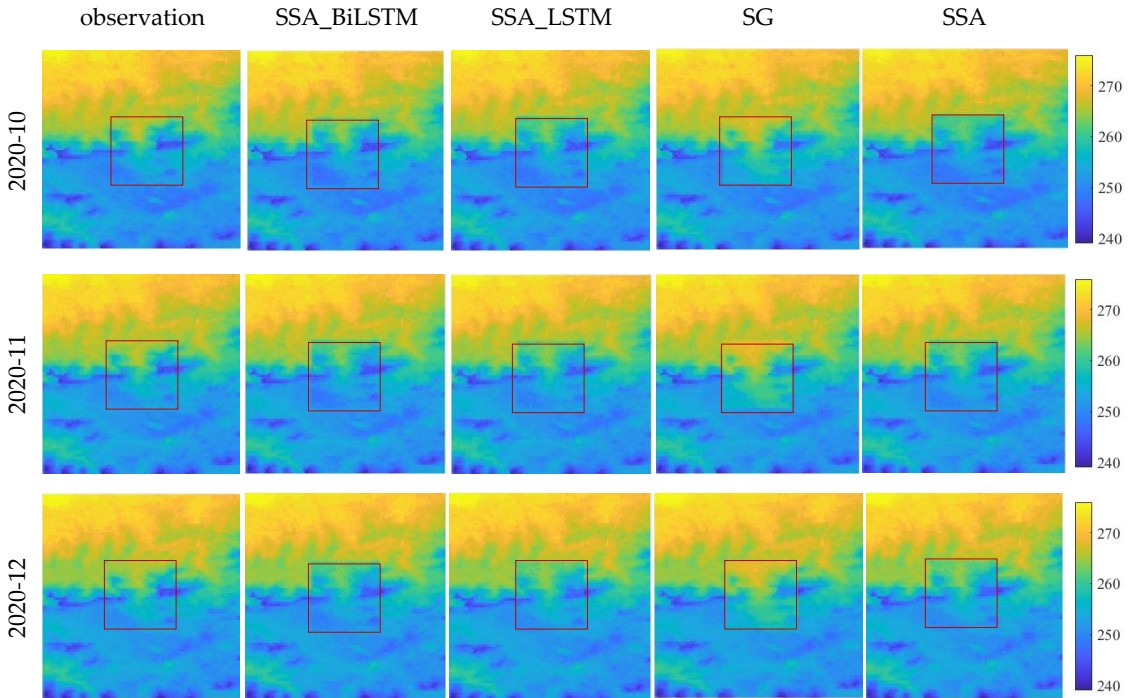

**Figure 8.** Comparison between the images reconstructed using different methods. The red blocks in the charts above represent the areas where the missing data are reconstructed. 2020-10, 2020-11 and 2020-12 represent issues 10, 11, and 12 of 2020, respectively.

The differences in the data reconstructed using different LST reconstruction methods and the original data were treated to demonstrate the performance of the proposed method in LST reconstruction in a more visual way. The results are shown in Figure 9. It can be seen from Figure 9 that the proposed method is accurate in LST reconstruction, while the images reconstructed using the method based on SG filter deviate greatly from the original images.

In order to verify the regional applicability of the proposed method, Wenchuan in the Sichuan province was selected as the validation area. A small range of $50 \times 50$ pixels were selected from the Wenchuan area, and all missing pixels in 2020 were reconstructed according to the method in this paper. The comparison effect before and after reconstruction is shown in Figure 10. The figure shows the reconstruction effect of this method on this region in different seasons. It can be seen that the method in this paper can achieve the reconstruction of a large number of missing pixels, and the reconstructed images are complete.

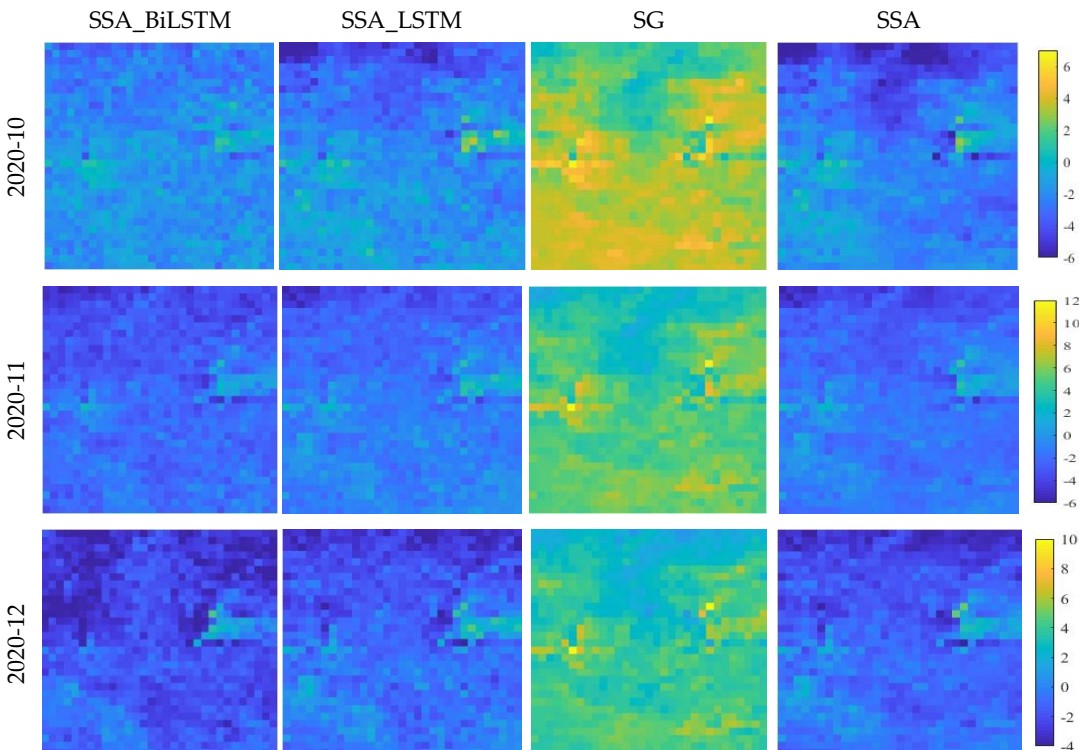

**Figure 9.** Comparison of images reconstructed using the proposed method and other LST reconstruction methods.

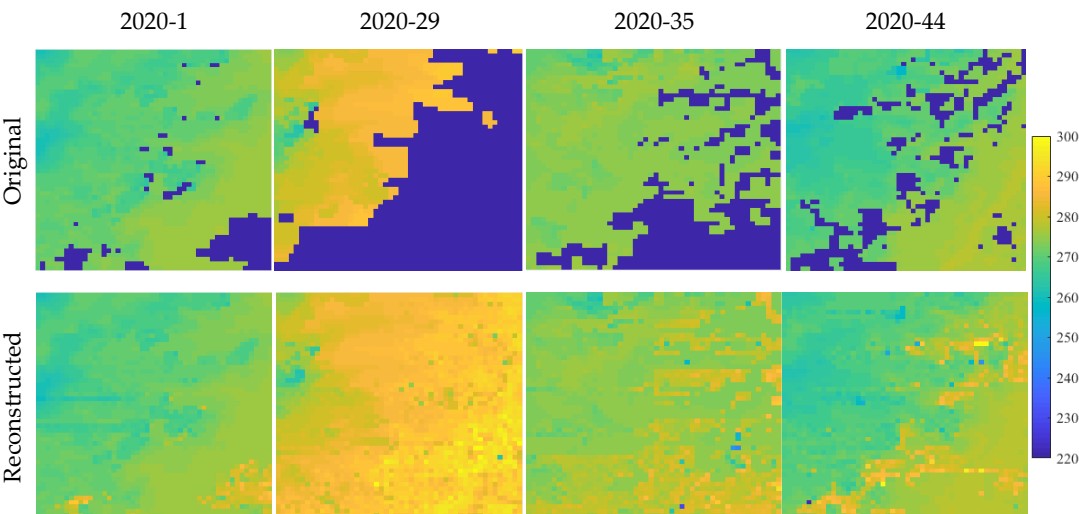

**Figure 10.** Comparison of the effect before and after reconstruction in the small area of Wenchuan, Sichuan province. The dark blue in the original image shows the missing areas.

### 4.3. Limitations of the Proposed Method

Although the proposed method can achieve relatively high accuracy in LST reconstruction when there are a large number of missing values in time series data, it has certain limitations. Firstly, the method in this paper lacks the use of spatial information. When further research is conducted in the future, consideration can be given to establishing a reconstruction model that combines a convolutional neural network with the predictive model to identify the spatial and temporal features of LST and achieve higher accuracy in LST reconstruction. Secondly, due to the significant impact of abrupt changes such as changes in weather conditions on LST reconstruction, the performance of the proposed method in reconstructing some pixels is unsatisfactory. In subsequent LST reconstruction

efforts, more attention should be paid to data reconstruction problems caused by changes in weather conditions.

## 5. Conclusions

The large number of missing values in MODIS LST data restricts the use of such data. SSA-BiLSTM, an LST reconstruction method combing data decomposition with data prediction, is proposed to obtain spatially and temporally continuous LST data. This method consists of two major processes, namely, rough LST reconstruction based on the trend features of the data extracted using the SSA model and refined LST reconstruction based on the short-term features of the data learned by BiLSTM model.

A comparative analysis of the four methods mentioned in this paper is performed through "removal–reconstruction–comparison" using RMSE, $R^2$, and MAPE based on remote sensing data and measured data. Experimental results show that when the missing rate is high, the deviations of data reconstructed using the methods based on SG filter and SSA are great, and the stability of reconstructed data is relatively low. Hybrid models based on data decomposition perform better than single models in LST reconstruction. The SSA-BiLSTM model is more accurate than the SSA-LSTM in LST reconstruction, indicating that compared with the latter, the former can consider the features of the entire time series data more adequately and perform better in predicting unknown data.

**Author Contributions:** J.C. planning and experimental design; M.Z. program design and manuscript preparation; D.S. analysis of results and revision of the manuscript revision; X.S. and B.W. review and revision of the manuscript. All authors have participated in discussions and revisions of the manuscript. All authors have read and agreed to the published version of the manuscript.

**Funding:** This study is supported in part by the National Key Research and Development Program of China [2019YFC1509202] and in part by the National Science Foundation of China [41772350, 61371189, and 41701513].

**Data Availability Statement:** All the data given in this paper are available at corresponding websites. For other results and data, a request for sharing may be sent to the corresponding author.

**Acknowledgments:** The MODIS data used in this paper are sourced from online databases provided by the Land Processes Distributed Active Archive Center (LP DAAC) of NASA and United States Geological Survey (USGS)/Earth Resources Observation and Science (EROS) Center downloadable at https://ladsweb.modaps.eosdis.nasa.gov/ (visited on 25 December 2021). Surface observation data are available at http://www.cma.gov.cn/ (visited on 25 December 2021).

**Conflicts of Interest:** There is no conflict of interest among the authors of this paper.

## Abbreviations

The following abbreviations are used in this manuscript:

| | |
|---|---|
| BiLSTM | Bidirectional Long Short-Term Memory |
| EMD | Empirical Mode Decomposition |
| LST | Land Surface Temperature |
| LSTM | Long Short-Term Memory |
| MAPE | Mean Absolute Percentage Error |
| MODIS | Moderate Resolution Imaging Spectroradiometer |
| MRT | MODIS Projection Tool |
| $R^2$ | Correlation Coefficient |
| RMSE | Root Mean Square error |
| RNN | Recurrent Neural Network |
| SG | Savitzky Golay |
| SSA | Singular Spectrum Analysis |

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
