# Peer review of "MODIS Land Surface Temperature Product Reconstruction Based on the SSA-BiLSTM Model"

_remotesensing, doi:10.3390/rs14040958_

Round 1
Reviewer 1 Report
The manuscript deals with the development of a new methodology to fill the gaps in LST MODIS data due to cloudy scenes. Even though the approach can be considered valid to reconstruct the missed LST values, some experiments and results need to be clarified and improved.
After careful consideration I reconsider the paper after major revision: the results have to be improved together with the general presentation of the paper.
In the following the details:
- Lines 86-87: Sentence repetition ('for most of the methods');
- Section 2.1: the study area is located in a region with very little precipitation. I think that the used approach has to be validated also in other different areas to have a general purpose;
- Section 2.2.1: the authors used 8-days MODIS data. They have to justify this choice (e.g. why not use daily measurements);
- Lines 137-139: weather stations were available to compare the results, but more information are necessary about these measurements, suche as the accuracy, the resolution, etc.
- Figure 1: specify this plot represents MODIS LST in the caption;
- Equation 1: specify the meaning of subscript 'T';
- Table 1: specify the difference between 'data1' and 'data2'. Furthermore in this table the R2 are reported. I am surprised that as the items increase, the R2 decreases for the case of 30, 40 and 50% of missing rate, which also in conflict with the assertion done in lines 182-184. Moreover it is not clear the chosen number of items in the used approach.
- Lines 236-238: Put the sentence after the explanation of Eq. 7-9;
- Equations 7-9: specify the meaning of all parameters included in these equations, such as φ, b, w, etc.
- Subsection 3.2.1 needs to be improved; it is very difficult to read and to understand how the refined LST reconstruction method has been implemented (e.g., the initial parameters could be placed in a table, etc.). Furthermore the authors must justify the choice of two iterations in the process (has an iteration test been performed?);
- Lines 311-313: I believe the authors meant the opposite: as missing rate decreases, the correlation increases;
- Figures 6-7: the R2 values reported in Figure 6 do not reflect those in Fig.7. The authors choice a group of data to plot Figure 7, without specifying the used criteria to select this group and the reason to not represent all data information;
- Lines 333-335: the explanation of the results is inverted. Please fix it.
- Lines 328-339: the authors have to give more detailed information such as the number of measurements provided by the weather stations used for the comparison, etc.
Reviewer 2 Report
The paper proposes a novel methodology to cope with missing LST values from MODIS instrument. The topic addressed is generally very important, as many satellite data suffer from discontinuity e.g. in presence of clouds or for technical problems. However, the description of the method and of the results should be improved.
Hereafter, specific comments.
- In the Introduction (page 1), when a word finishes in a newline, the hyphen is missing.
-
L 134: MODIS acronym already specified
-
L139 and Figure 1: it seems from the Figure (at least in April) that the stations are located where almost no MODIS data are available. In this case, the statistic may suffer when performing comparisons.
-
Figure 3: The y title is superimposed to the y values in the exes. The plot is not totally clear. Its readability may improve e.g. showing the differences w.r.t. the original value.
-
Table 1: It is not clear to me what data1 and data2 represent, it should be explained better both in the Table caption and in the text. It also seems that increasing the items from 15 to 20 there is no gain (or a small degradation of the fit). Could you comment on that?
-
L249. How are these numbers chosen? Did you test other configurations? May the best configuration depend on some parameters, e.g. the dimensions of the sample used?
-
In the 4.1 "Quantitative analysis" paragraph on page 9, when a word finishes in a newline, the hyphen is missing.
-
L 307: The symbol to denote kelvin should be written uppercase
-
Figure 6: It would also be better to put the unity of measure in the X axes and not in the title. The numbers inside the plots are also not well aligned (both vertically and horizontally) and therefore difficult to be read (some of them are too left, e.g. in R2 30% SSA-LSTM or missing e.g. MAPE SG 40%). Also, the background colours of the bar plots are not consistent between different plots (e.g. in R2 20% SG is in pink and R2 10% SG is in grey).
-
Figure 7: The y axes titles are superimposed to the y tick marks. The axes may also have different ranges between different plots. Please consider to uniform all the scales.
-
L335. Do you have also in this case, the coefficient of correlation for the other methods? It may also be worth adding the relative plots in Figure 8.
-
Figure 8: The y axes titles are superimposed to the y tick marks. Also, in this case, the axes have different ranges between the two plots. Please consider uniform all the scales.
-
L346: on which region, always the region selected in Figure 1?
-
L348: do you mean 40%?
-
L358 and Figure 9: Could you comment also on the other methods, are the differences expected and why e.g. for SG they are mostly positive? It also seems that the performances change a little bit in function of the month analyzed. It seems e.g. that in December the performances of the SSA_BiLSTM method worsen.
-
L369: It is not clear to me how is the spatial information used in the proposed method. Please clarify.
Round 2
Reviewer 1 Report
The authors performed the suggested corrections/comments by improving the quality of the paper considerably. Thus I accept the manuscript in present form.
Reviewer 2 Report
The Authors have addressed all of my concerns with the original manuscript. Therefore, I suggest publishing the manuscript in its present form.